# Mulched Drip Fertigation with Growth Inhibitors Reduces Bundle-Sheath Cell Leakage and Improves Photosynthesis Capacity and Barley Production in Semi-Arid Regions

**DOI:** 10.3390/plants13020239

**Published:** 2024-01-15

**Authors:** Yinping Xu, Jianhua Liu, Cheng Ren, Xiaoxia Niu, Tinghong Zhang, Kecang Huo

**Affiliations:** Institute of Industrial Crops and Malting Barley, Gansu Academy of Agricultural Sciences, Lanzhou 730070, China; yinping_xu@yahoo.com (Y.X.); jianhua_liu@yahoo.com (J.L.); chengren@yahoo.com (C.R.); xiaoxia_niu22@yahoo.com (X.N.); tinghong_zhang22@yahoo.com (T.Z.)

**Keywords:** mulched drip fertigation, bundle-sheath cell leakage, inhibitor strategies, photosynthetic capacity, ^13^C carbon isotope, barley productivity

## Abstract

A better understanding of the factors that reduce bundle-sheath cell leakage to CO_2_ (Փ), enhance 13C carbon isotope discrimination, and enhance the photosynthetic capacity of barley leaves will be useful to develop a nutrient- and water-saving strategy for dry-land farming systems. Therefore, barley plants were exposed to a novel nitrification inhibitor (NI) (3,4-dimethyl-1H-pyrazol-1-yl succinic acid) (DMPSA) and a urease inhibitor (UI) (N-butyl thiophosphorictriamide (NBPT)) with mulched drip fertigation treatments, which included HF (high-drip fertigation (370 mm) under a ridge furrow system), MF (75% of HF, moderate-drip fertigation under a ridge furrow system), LF (50% of HF, low-drip fertigation under a ridge furrow system), and TP (traditional planting with no inhibitors or drip fertigation strategies). The results indicated that the nitrification inhibitor combined with mulched drip fertigation significantly reduced bundle-sheath cell leakage to CO_2_ (Փ) as a result of increased soil water content; this was demonstrated by the light and CO_2_ response curves of the photosynthesis capacity (An), the apparent quantum efficiency (α), and the ^13^C-photosynthate distribution. In the inhibitor-based strategy, the use of the urease and nitrification inhibitors reduced Փ by 35% and 39% compared with TP. In the NI-HF strategy, it was found that barley could retain the maximum photosynthesis capacity by increasing the leaf area index (LAI), An, rubisco content, soluble protein, dry matter per plant, and productivity. The CO_2_ and light response curves were considerably improved in the NI-HF and NI-MF treatments due to a higher 13C carbon isotope (Δ‰), respiration rate (Rd), and Ci/Ca, therefore obtaining the minimum Փ value. With both inhibitors, there was a significant difference between HF and LF drip fertigation. The NI-MF treatment significantly increased the grain yield, total chlorophyll content, WUE, and NUE by 52%, 47%, 57%, and 45%, respectively. Collectively, the results suggest that the new nitrification inhibitor (DMPSA) with HF or MF mulched drip fertigation could be promoted in semi-arid regions in order to mitigate bundle-sheath cell leakage to CO_2_ (Փ), without negatively affecting barley production and leading to the nutrient and water use efficiency of barley.

## 1. Introduction

The shortage of irrigation water constrains crop production worldwide, and, with the projected climate change, its impacts will become more significant in the near future [1]. To cope with the water shortage, it is necessary to develop water-saving agriculture countermeasures, thereby producing more crops per drop. Rainwater is the main factor influencing barley production in the rain-fed areas of Northwest China [2]. Water shortages in semi-arid regions are also a severe ecological issue [3]. The shortage of irrigation water is the main constraint on worldwide crop productivity. Mulched drip fertigation with inhibitors is the best nutrient- and water-saving farming approach, widely used in dry-land farming systems [1]. Plastic film mulching is a technique that reduces soil water loss via evaporation, with the objectives of reducing water stress and improving crop production [4]. As the global population continues to grow, the demand for food is increasing, and the use of fertilizers is also increasing in farming systems [5]. Due to the high demand for water and fertilizers, especially nitrogen fertilizers, barley cultivation may result in significant nitrogen losses due to NH_3_ evaporation and nitrate leaching [6,7,8]. Finding the best management practices that lead to lower N losses while maintaining yields is therefore crucial in barley cropping areas to ensure both the economic and environmental sustainability of these agro-ecosystems.

Many physiological processes have an effect on crop yields, in which photosynthesis is the foundation of dry matter accumulation. Drought stress during the flowering stage has a deleterious effect on the photosynthetic capacity due to leaf senescence and photosynthesis reduction [9]. The senescence of leaves speeds up the degradation of stored dry matter, resulting in decreased Δ^13^C and negatively affecting the net photosynthetic rates of crops [10,11]. Understanding the effect of mulched drip fertigation with two novel nitrification and urease inhibitors on yields through photosynthesis will help to increase crop production [2]. Nitrification and urease inhibitors [12,13], water-saving drip fertigation approaches [14], and distributed nitrogen fertilization have been used to improve the management of the nitrogen supply and barley production [15], resulting in increased yields, reduced water consumption, and reduced agricultural costs [8]. Drip irrigation is often used for plant production in water-poor areas [14]. Mulched drip fertigation is combined with inhibitors to achieve the integrated fertigation of water and fertilizers, save irrigation water, reduce the evapotranspiration (ET) rate, and improve production and resource use efficiency [16,17]. Therefore, it is necessary to study the effects of nitrification and urease inhibitors on photosynthetic capacity and water use efficiency and the impact of drip irrigation on the Δ^13^C and barley yield.

Basic photosynthesis parameters are associated with leaf chlorophyll content and the bundle-sheath cell leakiness of CO_2_ (Փ). There is a considerable relationship between the leaf photosynthesis capacity and chlorophyll content [18]. However, it remains unclear how various mulched drip fertigation treatments with nitrification and urease inhibitors affect this relationship. Moreover, the carbon assimilation rates of barley leaves may be considerably improved by slight improvements in the soil water content [19]. The use of urease inhibitors is an effective strategy to mitigate Փ [20]. Nitrification inhibitors have been defined as suitable tools to improve the nitrogen utilization efficiency, thereby reducing nitrogen loss [21,22], which can also improve photosynthesis, the chlorophyl content, and the barley yield [13]. In addition, in C_4_ crops, the ratio of the carboxylation rate of phosphoenolpyruvate (PEP) to the decarboxylation acid rate of C_4_ to the bundle-sheath leakage of the CO_2_ rate is in equilibrium, maximizing the C_4_ photosynthesis capacity [23,24], indicating a negative correlation between PEP and photosynthesis [25,26]. Research on C_4_ grasses and crops has shown that Փ may be considerably influenced by environmental stress, such as rainwater stress, thus considerably increasing Փ [27,28]. However, so far, it is not clear whether it is influenced by nutrition and water-saving management strategies, such as mulched drip irrigation using nitrification and urease inhibitors [29].

In view of these challenges, there is limited research on the synergistic influence of urease (NBPT) and nitrification (DMPSA) inhibitors in mulched drip fertigation strategies. Mulched drip fertigation is a modern and innovative irrigation method that uses a drip fertigation system to apply fertilizer and irrigation water together, with the aim of improving the photosynthetic CO_2_ and light response curves, carbon isotope discrimination, nitrogen use efficiency (NUE), and water use efficiency (WUE); reducing CO_2_ sheath cell leakage (Փ) and surface evaporation; and increasing barley production. Still, it is unknown whether Δ^13^C is affected by water-saving agricultural strategies, such as mulched drip fertigation with nitrification and urease inhibitors, where the barley crop is exposed to moderate water stress. The specific aims of this study were to reveal the effect of mulched drip fertigation on physiological processes, CO_2_ sheath cell leakage (Փ), and production and to improve the photosynthesis capacity of barley using two novel nitrification and urease inhibitors. These mechanisms are examined in semi-arid areas where the photosynthesis capacity of barley leaves and Փ are affected by various nutrient- and water-saving agricultural strategies.

## 2. Results

### 2.1. Dry Matter Partitioning and ^13^C-Photosynthates in Different Organs

The synergistic influence of the urease (NBPT) and nitrification (DMPSA) inhibitors under the mulched drip fertigation strategies considerably affected dry matter partitioning to various organs under different growth stages (Table 1). At the maturity stage, the highest dry matter partitioning plant^−1^ was noted under the NI-HF treatment. There were significant effects of the mulched drip fertigation strategies on the amount of dry matter (g plant^−1^) at various growth stages of barley. On average, the data showed that the dry matter partitioning (g plant^−1^) was higher for NI-HF, and the lowest was noted with the TP planting. Compared with UI-HF, the dry matter (g plant^−1^) was considerably improved by 25.4% in the NI-HF treatment and 23.2% in NI-MF, whereas, in NI-HF, it was enhanced by 71.6%, compared with TP planting at the maturity stage.

The application of urease (NBPT) and nitrification (DMPSA) inhibitors under mulched drip irrigation changed the distribution of ^13^C photosynthetic patterns among several plant organs (Table 2). During physiological maturity, the enrichment rate of ^13^C is relatively high in the stems, grains, spikes, and leaves of barley crops. The average data show that under NI-HF treatment, the distribution of ^13^C photosynthesis in grains was the highest, followed by UI-HF treatment, and the distribution was the smallest under TP treatment. Compared with the UI-HF treatment, the distribution of ^13^C photosynthesis in the grains of NI-HF treatment and NI-MF treatment significantly increased by 10.1% and 4.2%, respectively, while that in NI-HF treatment increased by 30.0% compared to TP treatment. In addition, traditional planting and no drip irrigation and fertilization significantly reduced the photosynthetic distribution pattern of ^13^C in the leaves and stems, while mulched drip irrigation and fertilization under urease (NBPT) and nitrification (DMPSA) inhibitors considerably enhanced the ^13^C in the stems and spikes.

### 2.2. C_i_/C_a_ Ratio, Δ^13^C, and Փ

Under both urease (NBPT) and nitrification (DMPSA) inhibitors with mulched drip fertigation, the C_i_/C_a_ concentration ratio improved considerably (Figure 1). In the NI-HF treatment, the C_i_/C_a_ was considerably improved by 14.8% compared to TP treatment. The mean of the two years’ data indicated that the C_i_/C_a_ of the NI-MF and UI-HF treatments was enhanced by 7.7% and 11.1% compared to TP treatment. The Ci/Ca of the NI-LF treatment was improved considerably by 7.2%, and that in UI-MF was improved by 6.5%, compared with the TP treatment. Compared with the TP treatment, in the NI-HF treatment, (Δ^13^C) carbon isotope discrimination was considerably enhanced by 29.1%. Compared with TP treatment, the NI-HF, NI-MF, NI-LF, UI-HF, UI-MF, and UI-LF treatments significantly increased Δ^13^C by 29.1%, 20.4%, 18.2%, 22.5%, 18.9%, and 12.7%, respectively. Under the TP treatment, the leakage of bundle-sheath cells to CO_2_ (Փ) was 35.2% higher compared with NI-HF. Compared with TP, the NI-HF, NI-MF, NI-LF, UI-HF, UI-MF, and UI-LF treatments significantly decreased Փ by 35.2%, 22.8%, 22.9%, 38.7%, 29.8%, and 11.0%, respectively.

### 2.3. Respiration Rate and Apparent Quantum Efficiency (α)

For the apparent quantum efficiency (α), in the research years of 2020–2021 and 2021–2022, there were considerable changes under mulched drip irrigation combined with urease (NBPT) and nitrification (DMPSA) inhibitor strategies (Table 3). Compared to TP treatment, in the NI-HF and UI-HF treatments, the α increased by 17.1% and 12.6%, respectively, while that in the NI-MF treatment increased by 9.9%. Compared to TP treatment, at the grain filling stage, the average α values of the NI-HF and UI-HF treatments increased by 12.8% and 8.8%, respectively. The two-year average data show that under the NI-HF and TP treatments at different growth stages, the α also decreased from 0.056 to 0.044 and from 0.046 to 0.039, respectively. Compared with TP treatment, the NI-HF treatment increased the average respiratory rate (Rd) by 33.6%, 24.0%, and 20.5%. Compared with TP treatment, the mean Rd of the NI-HF and UI-HF treatments in the grain filling stage increased by 42.0% and 31.6%.

### 2.4. Soil Water Content (SWC) and Chlorophyll Content

The mulched drip fertigation strategies had significant effects on rainfall collection and enhanced SWC at the 0–200 cm layers compared to traditional planting with no inhibitors and drip fertigation (Figure 2). The SWC at 0–200 cm under the NI-HF treatment was significantly different at various growth stages. At the flowering and grain filling stages, the SWC at the 0–200 cm depth improved considerably under both urease (NBPT) and nitrification (DMPSA) inhibitors with mulched drip fertigation, while, under the UI-LF and TP treatments, the SWC gradually decreased as compared to the rest of the treatments. In addition, the nitrification (DMPSA) inhibitor with high drip mulched fertigation increased the SWC. Furthermore, at the flowering stage, the SWC under NI-HF and UI-HF was considerably improved by 55.1% and 49.4% compared to the TP treatment.

The mulched drip fertigation in combination with urease (NBPT) and nitrification (DMPSA) inhibitor strategies considerably improved the Chl ab content and chlorophyll stability index (CSI) of barley flag leaves. In both urease (NBPT) and nitrification (DMPSA) inhibitor with mulched drip fertigation strategies, the Chl ab content and CSI of barley improved with increasing mulched drip fertigation, while the CSI was considerably reduced with increasing mulched drip fertigation with both urease and nitrification inhibitors (Table 4). However, when mulched drip fertigation increased from moderate to high, there was no significant variance noted in the Chl ab content and CSI under both urease and nitrification inhibitors. The mean data from both years revealed that the NI-HF, NI-MF, NI-LF, UI-HF, UI-MF, and UI-LF treatments considerably improved the Chl ab content by 48.6%, 46.8%, 32.9%, 42.5%, 39.7%, and 20.3% compared to the TP treatment.

### 2.5. Soluble Protein, Rubisco Content, and Leaf Area Index (LAI)

In the two-year study, the soluble protein content in the leaves increased from 0 DAF to 40 DAF under the NI-HF and UI-HF treatments. The difference between the NI-HF and UI-HF treatments was significant from 0 to 32 DAF, while the change between NI-MF and TP was considerable between 0 and 40 DAF (Figure 3). In addition, from 0 to 40 DAF, the NI-HF and NI-MF treatments led to higher rubisco content in the leaves compared to all other treatments, and the NI-HF treatment showed the greatest improvement during the two-year study period (Figure 4). These results showed that, after flowering, mulching drip irrigation with urease (NBPT) and nitrification (DMPSA) inhibitors increased the content of soluble protein and rubisco in leaf photosynthesis, which was helpful for the accumulation of photosynthesis products and improved the photosynthetic capacity.

Under different strategies of mulched drip irrigation combined with urease (NBPT) and nitrification (DMPSA) inhibitors, the LAI from 2020 to 2022 was significantly higher (Figure 5). Under NI-HF, there were significant variations in the LAI of barley crops at various growth stages. The change in LAI started at 60DAP and reached its maximum at 100 DAP; it then gradually decreased at 100–140 DAP due to leaf senescence, indicating that early leaf senescence occurs due to inadequate rainwater. Under the action of urease and nitrification inhibitors, except for 30 DAP, different mulching drip irrigation fertilization strategies had a significant impact on the LAI. For example, the NI-HF treatment increased the LAI on different days after planting. However, the difference between the NI-MF and UI-HF treatments was not significant.

### 2.6. Photosynthesis Light and CO_2_ Response Curve

The light response curve of barley leaves under the combined action of various mulched drip irrigation fertilization and urease (NBPT) and nitrification (DMPSA) inhibitor strategies is shown in Figure 6. The photosynthetic response curve indicates the relationships between important photosynthetic parameters, such as *A_n_*. The light response curve shows that, due to the different mulched drip irrigation conditions with different combinations of urease and nitrification inhibitors, the dynamic changes in the number of standard stems of *A_n_* during barley flowering increased (Figure 7). Under different urease and nitrification inhibitor strategies, the *A_n_* of the NI-HF and UI-HF treatments was significantly higher as compared with the rest of the treatments. Compared with the UI-HF and UI-MF treatments, a higher *A_n_* was recorded under the NI-HF and NI-MF treatments. Similarly, when the number of standard rods exceeded 1600 µmol m^−2^ s^−1^, the Pn curve became constant. In descending order, the *A_n_* at various PAR levels in the different treatments was as follows: NI-HF > UI-HF > NI-MF > UI-MF > NI-LF > UI-LF and TP treatments.

We used the C_4_ photosynthetic model to simulate the CO_2_ response curve of photosynthesis at the flowering stage of barley flag leaves. The An CO_2_ response curve indicates that, at the flowering period of barley, with an increase in C_i_, drip irrigation fertilization combined with urease and nitrification inhibitor strategies under various mulches, the An rate was significantly increased (Figure 7). When observed under the nitrification inhibitor strategy, the *A_n_* content in flag leaves treated with NI-HF was the maximum as compared with the rest of the treatments. Compared with the TP treatment, NI-HF and UI-HF indicated the maximum *A_n_*. Likewise, when C_i_ exceeded 600 µmol mol^−1^, the *A_n_* curve became constant and saturated.

### 2.7. Resource Use Efficiency and Barley Production

The field study showed that the grain yield, WUE, and NUE of barley were significantly affected by the various mulched drip fertigation strategies in combination with urease (NBPT) and nitrification (DMPSA) inhibitors (Table 4). As the mulched drip fertigation increased, the grain yield considerably improved, but the differences were non-significant when the irrigation exceeded moderate mulched drip fertigation under both urease (NBPT) and nitrification (DMPSA) inhibitors. In addition, mulched drip fertigation irrigation combined with DMPSA considerably enhanced the NUE. Compared with UI-LF, the NUE of the NI-HF, NI-MF, NI-LF, UI-HF, and UI-MF treatments increased by 51.3%, 45.0%, 28.6%, 39.8%, and 27.0%. Meanwhile, the WUE in the NI-HF, NI-MF, NI-LF, UI-HF, UI-MF, and UI-LF treatments was increased by 59.3%, 56.9%, 44.0%, 48.4%, 44.8%, and 27.7% when compared with TP treatment. The application of the DMPSA inhibitor had the maximum effect on the WUE and NUE of barley. The grain yield in the NI-HF, NI-MF, NI-LF, UI-HF, UI-MF, and UI-LF treatments was increased by 55.8%, 51.5%, 34.5%, 48.7%, 42.1%, and 21.9% compared with TP treatment.

Financial profit is one of the most valuable estimation indexes for the management productivity of barley. The total costs of the different treatments had clear variations due to the use of plastic films, growth inhibitors, and irrigation (Table 5). The main output value for the barley crop was the grain yield. The output value under NI-HF was significantly higher than that of the rest of the treatments. The net incomes with the NI-HF, NI-MF, NI-LF, UI-HF, UI-MF, UI-LF, and TP treatments were 16,897, 14,895, 9295, 13,537, 11,295, 6655, and 4965 (Yuan ha^−1^). Among the various treatments, the net profit of NI-HF was the highest (16897 CNY ha^−1^).

## 3. Discussion

Due to water shortages, barley production can be limited by poor photosynthesis in rain-fed regions [30,31]. Earlier research works revealed that carbon isotope discrimination (Δ^13^C) plays a vital role in increasing the photosynthesis capacity and reducing the cell leakage of a bundle sheath to CO_2_ [23,32]. Compared with UI-HF, the dry matter (g plant^−1^) was considerably improved by 25.4% in the NI-HF treatment and 23.2% in NI-MF, whereas the NI-HF treatment enhanced it by 71.6% compared with TP planting at the maturity stage. In addition, traditional planting and drip irrigation and fertilization significantly reduced the photosynthetic distribution pattern of ^13^C in the leaves and stems, while mulched drip irrigation and fertilization under urease (NBPT) and nitrification (DMPSA) inhibitors considerably enhanced the ^13^C distribution in the spikes and stems. Earlier research has also reported similar results [33,34,35]. The timing of deficit irrigation can significantly improve the chlorophyll ab content and photosynthesis capacity and delay leaf senescence [10]. Compared to TP treatment, in the NI-HF treatment, the (Δ^13^C) carbon isotope identification was significantly improved by 29.1%. Under TP treatment, bundle sheath cell leakage was sensitive to CO_2_ (Փ) and was 35.2% higher than in the NI-HF treatment. The barley leaf carbon Δ^13^C can reveal the ecological effects of the barley crop Pn [36,37]. Ali et al. [1] reported that Δ^13^C was changed with the influence of a greater ρ and was considerably enhanced under the CF rather than the RB planting technique. Under the NI-HF treatment, Ci/Ca was considerably enhanced by 14.8% compared to the TP treatment. The Ci/Ca of the NI-MF and UI-HF treatments increased by 7.7% and 11.1%, respectively, compared to the TP treatment. The study by Wang et al. [32] showed that the maximum Ci/Ca value can lead to a significant increase in the Δ^13^C value. Thomas et al. [38] also found that, compared to DI treatment, PRI treatment had a significantly better Δ13C value. In the present research, the estimations of Δ^13^C and C_i_/C_a_ are similar to those of other C_4_ crops [39,40].

The barley yield is related to the photosynthetic capacity [41] and decreases under rain-fed agricultural systems Փ [38]. Compared to TP treatment, the average α of the NI-HF treatment and UI-HF treatment increased by 17.1% and 12.6%, respectively, while that of the NI-MF treatment increased by 9.9%. The two-year average data show that under the NI-HF and TP treatments at different growth stages, α decreased from 0.056 to 0.044 and from 0.046 to 0.039, respectively. Compared with TP treatment, the NI-HF treatment increased the mean Rd by 33.6%, 24.0%, and 20.5%, respectively. Mulching coverage leads to a rise in barley production, which is related to the *A_n_*, Rd, and α values [29]. Greater P_n_ and α values and a lower Փ are possible causes. Water scarcity decreases CO_2_ availability and stomatal conductance, thus decreasing the *A_n_* and resource use efficiency [42,43]. Numerous studies have confirmed that DMPSA inhibitors can improve the photosynthesis, Rd, and α values [44,45]. In irrigation experiments conducted in the same experimental area, Ali et al. [1] found that the impact of NBPT on the soluble protein and rubisco content was largely influenced by irrigation management in the weeks after fertilization, and the impact was negligible at the highest irrigation frequency. In this two-year study, the soluble protein content in the leaves increased from 0 DAF to 40 DAF under the NI-HF and UI-HF treatments. The difference between the NI-HF and UI-HF treatments was significant from 0 to 32 DAF, while the change between NI-MF and TP was considerable between 0 and 40 DAF. The rubisco enzyme in the Calvin cycle accounts for 50% of the soluble protein of leaves [46,47], so it is the most obvious target to improve the photosynthesis of barley [48]. These results showed that after flowering, the mulched drip irrigation with NBPT and DMPSA inhibitors increased the rubisco content under leaf photosynthesis, which was conducive to the accumulation of photosynthetic products and the improvement of photosynthesis.

Under the urease (NBPT) and nitrification (DMPSA) inhibitors and film drip irrigation strategies, the chlorophyll ab content and CSI in barley flag leaves increased with the increase in film drip irrigation fertilization, while the CSI decreased significantly with the increase in film irrigation under different urease and nitrification inhibitors. The amount and duration of irrigation considerably affects leaf senescence and increases the chlorophyll content and Pn [49,50]. However, when the level of covered drip irrigation fertilization increased from moderate to high, there was insignificant variance in the Chl ab content and CSI under the action of the urease and nitrification inhibitors. Increasing photosynthesis is one of the essential strategies to increase grain yields and biomass [51,52]. Previous studies have reported that RF systems play a crucial role in improving the water use efficiency in dry-land agricultural systems [39,49]. In this study, in the flowering and filling stages, under the urease (NBPT) and nitrification (DMPSA) inhibitors and the mulched drip irrigation fertilization strategy, the SWC at a 0–200 cm depth significantly increased (*p* < 0.05), while, under UI-LF and TP, the SWC gradually decreased. The light response curve shows that under the combined action of urease and nitrification inhibitors, the An and PAR of barley during the flowering period increased under different mulching drip irrigation and fertilization conditions. Under different urease and nitrification inhibitor strategies, the *A_n_* of the NI-HF and UI-HF treatments was considerably greater. Compared with the UI-HF and UI-MF treatments, a higher *A_n_* was recorded under the NI-HF and NI-MF treatments. Similarly, when the PAR exceeded 1600 µmol m^−2^ s^−1^, the *A_n_* curve became constant. We used the photosynthesis model developed by [53,54] to simulate the photosynthesis CO_2_ response curve at the flowering stage for barley flag leaves. Compared with TP treatment, NI-HF and UI-HF yielded the maximum *A_n_* values. Likewise, when C_i_ exceeded 600 µmol mol^−1^, the *A_n_* curve became constant and saturated. When Ci was 400 µmol mol^−1^, the CO_2_ response curve of *A_n_* in the barley leaves was the maximum [50]. Higher Δ^13^C may also contribute to a higher P_n_ value, barley yield, and photosynthesis with low Փ values [11,54]. This is consistent with our research results.

With the increase in the fertilization amount under mulched drip irrigation, the grain yield significantly increases, but when the fertilization amount under drip irrigation exceeds the moderate level of NBPT and DMPSA inhibitors, the yield change is not significant. In addition, the combination of mulched drip irrigation and DMPSA irrigation significantly improves the NUE. Wang et al. [31] found that the slow release of a nitrogen fertilizer can promote the transfer of nitrogen to barley grains. The improvement in water and nutrient utilization efficiency makes drip irrigation fertilization a promising strategy to improve plant nutrition and reduce nutrient losses (such as N) [48,53,55]. The WUE in the NI-HF, NI-MF, NI-LF, UI-HF, UI-MF, and UI-LF treatments was increased by 59.3%, 56.9%, 44.0%, 48.4%, 44.8%, and 27.7% when compared with TP treatment. The application of the DMPSA inhibitor had the maximum effect on the WUE and NUE of barley. The grain yield for the NI-HF, NI-MF, NI-LF, UI-HF, UI-MF, and UI-LF treatments was 55.8%, 51.5%, 34.5%, 48.7%, 42.1%, and 21.9% higher compared with TP treatment. The impact of nitrogen on barley yields is indirectly generated by growth effects [52,56]. The interaction between deficit irrigation and inhibitors is a key factor affecting barley growth and yields [51,54,57]. Furthermore, urease inhibitors inhibit the hydrolysis rate of amide nitrogen, reduce the volatilization of NH_4_^+^, prolong nitrogen fertilizer utilization, and ensure a sufficient nutrient supply for yield formation in the later growth stage [1], ultimately improving the NUE, WUE, and barley yield.

## 4. Materials and Methods

### 4.1. Site Description

This field experiment research was conducted in Huangyang Town, Liangzhou District, Wuwei City, Agricultural Science, Gansu Province, in 2020–2021 and 2021–2022, located at a latitude of 37°30′ N and longitude of 103°50′ E. The annual sunshine duration was 2360–2920 h, the height above mean sea level was 1766 m, and the frost-free period was 135–150 days. The rainfall in 2020–2021 was 230 mm, and that in 2021–2022 was 200 mm. The monthly rainfall and temperature in 2020–2021 and 2021–2022, as well as the monthly average over 40 years (1980–2020), are shown in Figure 8. Table 6 provides the soil characteristics at a depth of 0–20 cm in farmland over the past two years.

### 4.2. Experimental Design

The on-site study was carried out using a random complete block design (RCBD) with four replicates, each covering an area of 60 square meters (20 × 3 m). The study consisted of seven treatments: NI-HF—nitrification inhibitor (3,4-dimethyl-1H-pyrazole-1-yl succinic acid) (DMPSA) (Zhongsheng Agricultural Science (Shandong) Fertilizer Co., Ltd., Liaocheng City, China) with (370 mm) high drip fertigation irrigation under a ridge furrow system; NI-MF—nitrification inhibitor DMPSA with 75% of H, moderate drip fertigation irrigation under a ridge furrow system; NI-LF—nitrification inhibitor DMPSA with 50% of H, low drip fertigation irrigation under a ridge furrow system; UI-HF—urease inhibitor (N-butyl thiophosphorictriamide) (NBPT) with (370 mm) high drip fertigation irrigation under a ridge furrow system; UI-MF—urease inhibitor NBPT with 75% of H, moderate drip fertigation irrigation under a ridge furrow system; UI-LF—urease inhibitor NBPT with 50% of H, low drip fertigation irrigation under a ridge furrow system; TP—traditional planting with no inhibitor and drip irrigation. Drip irrigation under a plastic film cover was used for fertigation and irrigation. Before sowing, a drip irrigation belt was laid and then the ridge furrow area was covered with plastic film mulching. Each plot was equipped with high-precision water meters (LXS-25, Ningbo, China), pressure gauges, and control valves to ensure accurate discharge and a stable pressure. All experimental plots were irrigated at 50 mm after sowing to achieve uniform germination. Throughout the growing season, single water applications (64, 45, and 27 mm) were applied for the H, M, and L treatments, totaling 5 times. The AZ34 barley variety was sown with a population of 2.25 × 10^6^ seeds ha^−1^. The seeds were sown on 5 October 2020 and 9 October 2021, with row spacing of 20 cm. Barley was harvested on 19 May 2021 and 21 May 2022. A large space of 0.8 m was reserved between the bordering plots to prevent the leakage of water and nutrients. In these two years, 40 kg N ha^−1^, 200 kg P_2_O_5_ ha^−1^, and 27.5 kg K ha^−1^ were applied before planting, and 160 and 40 kg ha^−1^ of N were applied through drip irrigation fertilization during the jointing and flowering stages.

### 4.3. Soil Water Content

The soil moisture content at different growth stages was measured. A TDR machine (Time-Domain Reflectometry, Trase System, Soil Moisture Equipment Corp., Germany) was used to record the soil depth at 0–200 cm, with intervals of 20 cm.

### 4.4. δ^13^C Isotope and Chlorophyll Content

In the mature stage, the biomass of leaves was used for δ^13^C measurement. A fine powder was prepared from dry leaf samples using the Isoprime 100 instrument (Isoprime 100%, Cheadle, UK) for ^13^C analysis.

The ^13^C composition (δ^13^C) was calculated as
(1)$$\delta^{13}C=1000\left[\frac{R_{\mathrm{sample}\;}-R_{\mathrm{standard}\;}}{R_{\mathrm{standard}\;}}\right]$$
where *R_standard_* and *R_sample_* are the Pee Dee Belemnite (PDB) standard and ^13^C/^12^C ratios of the dry leaf sample.

The carbon isotope Δ^13^C was calculated according to Farquhar et al. [28].
(2)$$\Delta^{13}C_{\mathrm{plant}\;}\left(‰\right)=\left(\frac{\delta^{13}C_{\mathrm{air}\;}\left(‰\right)-\delta^{13}C_{\mathrm{plant}\;}\left(‰\right)}{\frac{\delta^{13}C_{\mathrm{plant}\;}\left(‰\right)}{1000}+1}\right)$$

In addition, five plant samples were obtained from the center of each plot at the maturity stage and subsequently divided into the ear leaf, stem (including sheath), other leaves, cob, ear bracts, tassel, and grain. All separated components were oven-dried at 80 °C to a constant weight. A sub-sample of 4 mg was used to determine the isotopic abundance. The total chlorophyll ab content of barley flag leaves was calculated based on the program and formula explained by von Camemmer and Furbank [29].

### 4.5. Leakiness of Bundle-Sheath Cells of CO_2_ (Փ)

Փ was calculated according to Farquhar et al. [28].
(3)$$\Phi =\frac{\Delta^{13}C-a+\left(a-b_{4}\right)C_{i}/C_{a}}{\left(b_{3}-s\right)C_{i}/C_{a}}$$

PEPC (2.2‰) is *b*_4_ (−5.7‰), the fractionation by rubisco is *b*_3_ (30‰), and the fractionation linked with the leakage of CO_2_ from the bundle sheath to the mesophyll is *s* (1.8‰); see Leegood [30].

### 4.6. Photosynthetic CO_2_ and Light Response Curve

A LI Cor LI-6400XT portable photosynthesis system (LI-6400XT, LI-Cor, Lincoln, NE, USA) was used to measure the photosynthetic CO_2_ response curve at a PPFD level of 1500 µmol m^−2^ s^−1^ during flowering. The CO_2_ levels were 300, 200, 100, 50, 400, 600, 800, 1200, 1600, and 2000 µmol CO_2_ mol^−1^, respectively. *A_n_* values were plotted against the intercellular CO_2_ concentrations (*C_i_*) to produce a response curve. According to Wang et al. [31], the CO_2_ response curve was calculated.
(4)$$A_{n}=a\left(1-e^{-bx}\right)+c$$
where *A_n_* is the net photosynthetic rate and *x* is *C_i_*.

The light response curves were measured during the same days on which the CO_2_ response curves were measured. The different PPFD levels were measured using the LI-Cor LI-6400XT portable photosynthesis system at a CO_2_ concentration of 400 µL L^−1^. PPFD levels were taken at 1800, 1600, 1400, 1000, 600, 400, 200, 100, 80, 60, 40, 20, and 0 µmol m^−2^ s^−1^. The light response curve was modeled by a non-rectangular hyperbolic model. Photosynthetic parameters derived from the light response curves were determined according to the method described by Ye [30].
(5)$$A_{n}=\frac{\alpha Q+A_{max}-\sqrt{{\left(\alpha Q+A_{max}\right)}^{2}-4\kappa \alpha QA_{max}}}{2\kappa }-R_{d}$$

Among them, *A_n_* is photosynthesis; *Q* is PPFD; *A*_max_ is the irradiance saturation rate of total photosynthesis; *R_d_* is the dark breathing rate; *a* is the maximum apparent quantum yield of CO_2_; J is a dimensionless convexity term (0 < j < 1). The slope of this line is the apparent quantum efficiency (α), and the absolute value of the intercept is the respiratory rate (*R_d_*).

### 4.7. Leaf Area Index (LAI) and Resource Use Efficiency

The LAI was calculated as
LAI = GLA × N/S (6)
where GLA is the green leaf area, N is the number of plants within a unit area of land, and S is the unit area of land.

The nitrogen use efficiency (NUE) was determined as below [33].
NUE = (NF − NC)/NA × 100 (7)
where NF is the N content in the fertilization plot, NC is the N content in the control plot, and NA is the nitrogen amount.
The WUE was calculated as WUE = Y/ET (8)

Water use efficiency (WUE), grain yield (Y), and evapotranspiration rate (ET).

The soluble protein content in leaves was measured according to Bradford’s [34] improved method. As mentioned earlier, the rubisco content in leaves was measured using SDS-PAGE [35]. Four rows from each plot were manually harvested to determine the grain yield.

### 4.8. Economic Analysis

The economic benefit for each plot was determined using the following equations:OV = Y_g_ × P_g_
IV = LC + PMC + MCC + SFC + IC + GI
O/I = OV/IV
NI = OV − IV
where OV: output value (CNY ha^−1^); Y_g_: grain yield; P_g_: local price of grain, respectively; IV: input value; LC: labor costs; PMC: plastic mulching costs; MCC: machine cultivation costs; SFC: seed and fertilizer costs; WC: water costs; IV: input value; O/I: output/input; NI: net income.

### 4.9. Statistical Analysis

The data were analyzed for each sampled event separately using the SPSS 22.0 software (SPSS Inc., Chicago, IL, USA) and plotted with Origin Pro 8.5 (Origin Lab Corporation, Northampton, MA, USA). Means and standard errors were calculated for individual measurements taken at each sampling date. To identify significant treatment effects, multiple comparisons were performed with the least significant difference (LSD) test, and the significance level was set at the 0.05 probability level.

## 5. Conclusions

Mulched drip fertigation with a new nitrification inhibitor (DMPSA) had a significant impact on the soil water content and bundle-sheath cell leakage to CO_2_ (Փ) and enhanced the ^13^C-photosynthate distribution in different organs, the photosynthetic capacity, the apparent quantum efficiency (α), and barley production. Under the inhibitor-based strategy, the use of urease and nitrification reduced Փ by 35% and 39% compared with TP. Under the NI-HF treatment, barley can maintain a high photosynthetic capacity at the flowering stage by improving the *A_n_*, LAI, soluble protein, rubisco content, dry matter per plant, and barley production. The light and CO_2_ response curves were significantly enhanced under the NI-HF or NI-MF treatments due to the higher ^13^C carbon isotope (Δ‰), respiration rate (Rd), and Ci/Ca as a result of the lower Փ compared to the rest of the treatments. Under both inhibitors, there was a significant difference between HF and LF drip fertigation. The NI-MF treatment significantly increased the grain yield, total Chl ab content, WUE, and NUE by 52%, 47%, 57%, and 45%, respectively. The new nitrification inhibitor (DMPSA) with HF or MF mulched drip fertigation must be assessed in future studies to confirm its potential to mitigate bundle-sheath leakiness to CO_2_ (Փ) without negatively affecting barley production and resource use efficiency.

## Figures and Tables

**Figure 1 plants-13-00239-f001:**
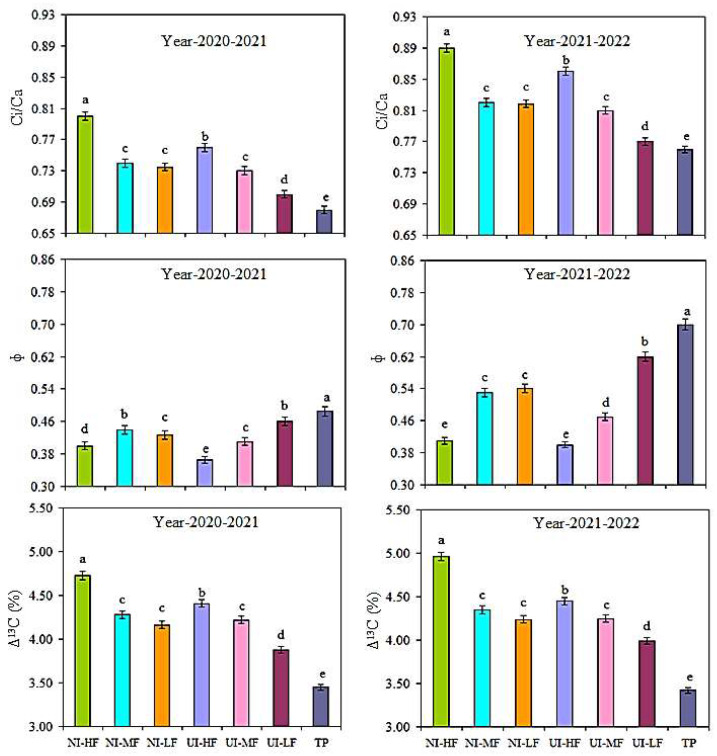
Effects of urease and nitrification inhibitor application with drip irrigation fertigation on the intercellular to ambient CO_2_ concentration ratio (Ci/Ca), bundle-sheath leakiness to CO_2_ (Փ), and carbon isotope discrimination (Δ^13^C) during the flowering stage. Vertical bars represent the (means ± SR) (n = 3), while lowercase letters represent LSD values.

**Figure 2 plants-13-00239-f002:**
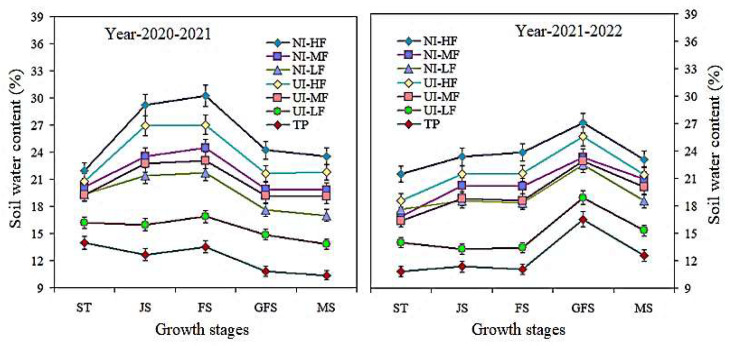
Dynamic changes in the soil water content (%) at different barley growth stages with 0–200 cm depth under different treatments during 2020–2022. Vertical bars represent the means ± SR (n = 3).

**Figure 3 plants-13-00239-f003:**
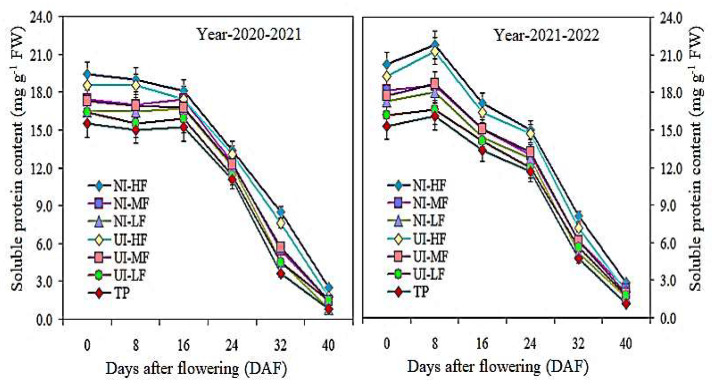
Effects of urease and nitrification inhibitor application with drip irrigation fertigation on the soluble protein content at different days after flowering. Vertical bars represent the means ± SR (n = 3).

**Figure 4 plants-13-00239-f004:**
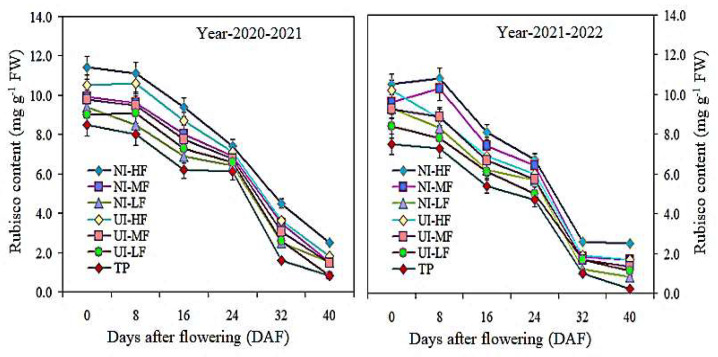
Effects of urease and nitrification inhibitor application with drip irrigation fertigation on the rubisco content at different days after flowering. Vertical bars represent the means ± SR (n = 3).

**Figure 5 plants-13-00239-f005:**
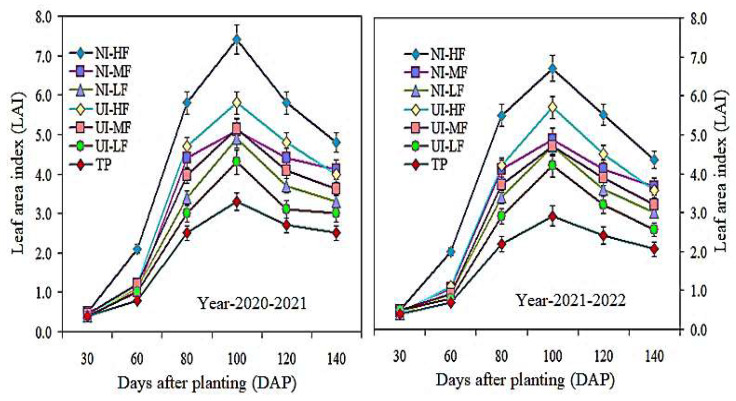
Dynamic changes in the leaf area index (LAI) at different barley growth stages under different treatments during 2020–2022. Vertical bars represent the means ± SR (n = 3).

**Figure 6 plants-13-00239-f006:**
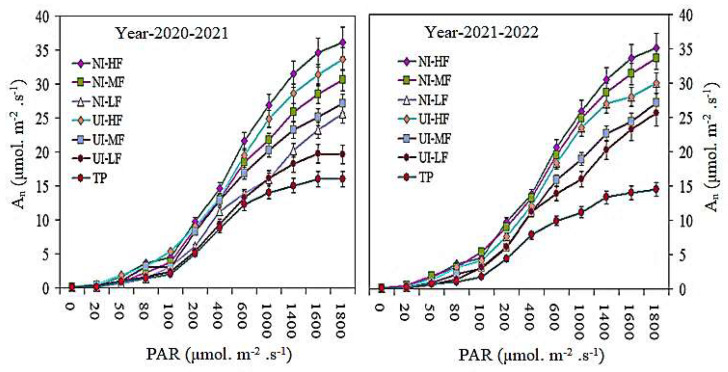
Dynamic photosynthetic light response curves of barley leaves exposed to urease and nitrification inhibitor application with drip irrigation fertigation during flowering stage (the measurements were made at a CO_2_ concentration of 400 µL L^−1^). Vertical bars represent the means ± SR (n = 3).

**Figure 7 plants-13-00239-f007:**
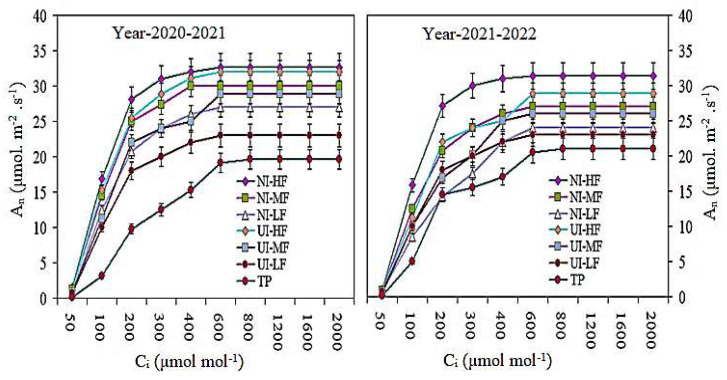
Photosynthetic CO_2_ response curves of barley leaves exposed to urease and nitrification inhibitor application with drip irrigation fertigation during flowering stage (measurements were made at a PPFD of 1500 µmol·m^−2^·s^−1^). Vertical bars represent the means ± SR (n = 3).

**Figure 8 plants-13-00239-f008:**
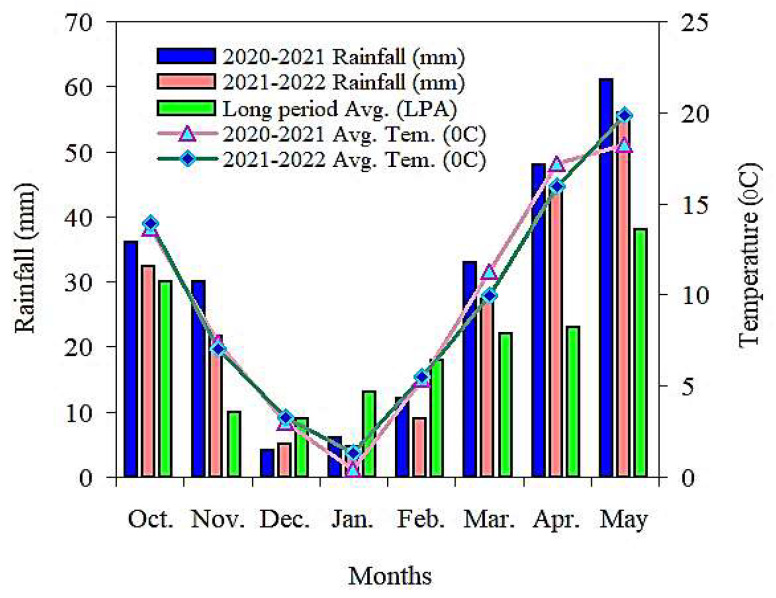
Monthly rainfall and temperature distribution during the barley–growing seasons.

**Table 1 plants-13-00239-t001:** Effects of different treatments on dry matter partitioning (g plant^−1^) in different organs (%) at physiological maturity of barley during 2020–2022.

Year	Treatment	Dry Matter (g plant^−1^)				
		TS	RWS	JS	FS	MS
2020–2021	NI-HF	0.45a	1.41a	5.26a	7.51a	10.45a
	NI-MF	0.40b	1.22b	4.32b	6.24b	8.78b
	NI-LF	0.33c	0.99c	2.35d	4.46d	6.26d
	UI-HF	0.41b	1.21b	4.09b	6.16b	8.76b
	UI-MF	0.34c	1.09c	3.24c	5.44c	7.14c
	UI-LF	0.27d	0.80d	2.04d	4.01d	4.45e
	TP	0.26d	0.68e	1.52e	3.81e	4.32e
2021–2022	NI-HF	0.90a	1.44a	3.20a	8.23a	15.00a
	NI-MF	0.89a	1.23c	2.27b	5.32b	12.54b
	NI-LF	0.70b	0.99d	1.99c	4.14c	8.31c
	UI-HF	0.75b	1.33b	2.80b	5.81b	11.53b
	UI-MF	0.57c	1.14c	2.15b	4.58c	10.16b
	UI-LF	0.46d	0.96d	1.55c	3.66d	5.45d
	TP	0.42d	0.43e	0.97d	2.55e	2.90e

NI-HF: nitrification inhibitor DMPSA with 370 mm high drip fertigation irrigation under ridge furrow system; NI-MF: nitrification inhibitor DMPSA with 75% of H, moderate drip fertigation irrigation under ridge furrow system; NI-LF: nitrification inhibitor DMPSA with 50% of H, low drip fertigation irrigation under ridge furrow system; UI-HF: urease inhibitor NBPT with 370 mm high drip fertigation irrigation under ridge furrow system; UI-MF: urease inhibitor NBPT with 75% of H, moderate drip fertigation irrigation under ridge furrow system; UI-LF: urease inhibitor NBPT with 50% of H, low drip fertigation irrigation under ridge furrow system; TP: traditional planting with no inhibitor and drip irrigation. TS: tillering stage; RWS: re-wintering stage; JS: jointing stage; FS: flowering stage; MS: maturity stage. Values are given as means, and different lowercase letters indicate significant differences at *p* ≤ 0.05 level in the same line (LSD test) (n = 3).

**Table 2 plants-13-00239-t002:** Effects of different treatments on 13C-photosynthates distribution in different organs (%) at physiological maturity of barley during 2020–2022.

Year	Treatment	^13^C-Photosynthate Distribution in Different Organs (%)			
		Stem	Leaves	Spike	Grain
2020–2021	NI-HF	48.9a	19.9a	1.8a	47.8a
	NI-MF	43.1b	19.6a	1.0a	42.2b
	NI-LF	38.3c	19.2a	1.0a	39.0c
	UI-HF	42.9b	19.7a	1.5a	44.6b
	UI-MF	40.7b	19.5a	1.2a	41.0b
	UI-LF	37.4c	19.8a	1.4a	36.6d
	TP	33.7d	18.7b	0.6b	35.6d
2021–2022	NI-HF	54.5a	24.0a	1.9a	47.3a
	NI-MF	52.0b	20.5b	1.8a	40.5b
	NI-LF	50.7c	17.9c	1.6a	35.7d
	UI-HF	52.3b	21.2b	1.6a	40.9b
	UI-MF	51.5b	19.5c	1.6a	38.3c
	UI-LF	50.1c	18.4c	1.4a	34.5d
	TP	49.4d	15.2d	1.4a	31.0e

NI-HF: nitrification inhibitor DMPSA with 370 mm high drip fertigation irrigation under ridge furrow system; NI-MF: nitrification inhibitor DMPSA with 75% of H, moderate drip fertigation irrigation under ridge furrow system; NI-LF: nitrification inhibitor DMPSA with 50% of H, low drip fertigation irrigation under ridge furrow system; UI-HF: urease inhibitor NBPT with 370 mm high drip fertigation irrigation under ridge furrow system; UI-MF: urease inhibitor NBPT with 75% of H, moderate drip fertigation irrigation under ridge furrow system; UI-LF: urease inhibitor NBPT with 50% of H, low drip fertigation irrigation under ridge furrow system; TP: traditional planting with no inhibitor and drip irrigation. Values are given as means, and different lowercase letters indicate significant differences at *p* ≤ 0.05 levels in the same line (LSD test) (n = 3).

**Table 3 plants-13-00239-t003:** Effects of different treatments on apparent quantum efficiency (α, dimensionless) and respiration rate (Rd, μmol m^−2^ s^−1^) at different growth stages of barley during 2020–2022.

Year	Treatment	Apparent Quantum Efficiency (α)		Respiration Rate (Rd, μmol m^−2^ s^−1^)	
		FS	GFS	FS	GFS
2020–2021	NI-HF	0.053a	0.047a	3.40a	3.70a
	NI-MF	0.051a	0.044b	3.00a	3.10a
	NI-LF	0.050b	0.044b	2.80b	2.75b
	UI-HF	0.052a	0.046a	2.75b	3.00a
	UI-MF	0.050b	0.044b	2.85b	2.50b
	UI-LF	0.048b	0.042b	2.50b	2.30b
	TP	0.046c	0.040c	2.30c	1.90c
2021–2022	NI-HF	0.058a	0.045a	4.00a	3.20a
	NI-MF	0.052b	0.040b	3.60b	2.80b
	NI-LF	0.047c	0.040b	3.05b	2.45b
	UI-HF	0.053b	0.043a	3.35b	2.85b
	UI-MF	0.050b	0.040b	3.20b	2.50b
	UI-LF	0.047c	0.038b	2.70c	2.65b
	TP	0.042d	0.036c	2.50c	2.10c

NI-HF: nitrification inhibitor DMPSA with 370 mm high drip fertigation irrigation under ridge furrow system; NI-MF: nitrification inhibitor DMPSA with 75% of H, moderate drip fertigation irrigation under ridge furrow system; NI-LF: nitrification inhibitor DMPSA with 50% of H, low drip fertigation irrigation under ridge furrow system; UI-HF: urease inhibitor NBPT with 370 mm high drip fertigation irrigation under ridge furrow system; UI-MF: urease inhibitor NBPT with 75% of H, moderate drip fertigation irrigation under ridge furrow system; UI-LF: urease inhibitor NBPT with 50% of H, low drip fertigation irrigation under ridge furrow system; TP: traditional planting with no inhibitor and drip irrigation. FS: flowering stage; GFS: grain filling stage. Values are given as means, and different lowercase letters indicate significant differences at *p* ≤ 0.05 level in the same line (LSD test) (n = 3).

**Table 4 plants-13-00239-t004:** Effects of different treatments on grain yield (t ha^−1^), total chlorophyll ab (mg g^−1^), and chlorophyll stability index (CSI, %) at average of three different growth stages (jointing, flowering, and grain filling stage), as well as nitrogen use efficiency and water use efficiency (WUE kg mm^−1^) of barley during 2020–2022.

Year	Treatment	Grain Yield (t ha^−1^)	Total Chl ab (mg g^−1^)	WUE (kg mm^−1^ ha^−1^)	CSI (%)	NUE (%)
2020–2021	NI-HF	10.3a	15.1a	27.0a	53.0c	36.3a
	NI-MF	9.3a	14.6a	25.3a	54.8c	30.5b
	NI-LF	6.5c	11.7c	18.7b	68.4b	21.7d
	UI-HF	8.6b	13.7b	20.9b	58.4c	29.3b
	UI-MF	7.3b	13.1b	18.9b	61.1c	26.0c
	UI-LF	5.3c	10.1c	13.8c	79.2a	15.7e
	TP	4.6d	8.0d	11.6c	-	-
2021–2022	NI-HF	9.9a	13.9a	30.0a	49.6d	44.0a
	NI-MF	9.1a	13.4a	28.5a	51.5c	40.6b
	NI-LF	7.1c	10.5b	22.7b	65.7b	33.0c
	UI-HF	8.8b	12.2b	24.1b	56.6c	35.6c
	UI-MF	8.1b	11.6b	23.1b	59.5c	27.5d
	UI-LF	6.1d	8.6c	18.3c	80.2a	23.4e
	TP	4.3e	6.9c	11.6d	-	-

NI-HF: nitrification inhibitor DMPSA with 370 mm high drip fertigation irrigation under ridge furrow system; NI-MF: nitrification inhibitor DMPSA with 75% of H, moderate drip fertigation irrigation under ridge furrow system; NI-LF: nitrification inhibitor DMPSA with 50% of H, low drip fertigation irrigation under ridge furrow system; UI-HF: urease inhibitor NBPT with 370 mm high drip fertigation irrigation under ridge furrow system; UI-MF: urease inhibitor NBPT with 75% of H, moderate drip fertigation irrigation under ridge furrow system; UI-LF: urease inhibitor NBPT with 50% of H, low drip fertigation irrigation under ridge furrow system; TP: traditional planting with no inhibitor and drip irrigation. Values are given as means, and different lowercase letters indicate significant differences at *p* ≤ 0.05 level in the same line (LSD test) (n = 3).

**Table 5 plants-13-00239-t005:** Average economic benefits in Chinese yuan (CNY ha^−1^) under different treatments.

Treatment	G.Y.	LC	PFC	MCC	SFC	IC	IV	OV	O/I	NI	NID
NI-HF	10.1	2627	710	1450	1967	638	7392	24,288	3.29	16,897	11,932
NI-MF	9.2	2627	710	1450	1967	479	7233	22,128	3.06	14,895	9930
NI-LF	6.8	2627	710	1450	1967	319	7073	16,368	2.31	9295	4330
UI-HF	8.7	2627	710	1450	1967	638	7392	20,928	2.83	13,537	8572
UI-MF	7.7	2627	710	1450	1967	479	7233	18,528	2.56	11,295	6330
UI-LF	5.7	2627	710	1450	1967	319	7073	13,728	1.94	6655	1690
TP	4.5	2627	0	1169	1967	0	5763	10,728	1.86	4965	-

Note: G.Y.: grain yield (t ha^−1^); LC: labor costs (Chinese yuan (CNY) ha^−1^); PFC: plastic film costs (CNY ha^−1^); MCC: machine cultivation costs (CNY ha^−1^); SFC: seed and fertilizer costs (CNY ha^−1^); IC: irrigation cost (CNY ha^−1^); IV: input value (CNY ha^−1^); OV: output value (CNY ha^−1^); O/I: output/input; NI: net income (CNY ha^−1^); NID: net income difference (CNY ha^−1^). Labor cost = 80 CNY per person day^−1^; plastic film cost = 12 CNY kg^−1^; barley grain price = 2.4 CNY kg^−1^.

**Table 6 plants-13-00239-t006:** The chemical properties of the soil layers at the experimental site (0–20 cm).

Year	pH (cm)	SOM (g kg^−1^)	TN (g kg^−1^)	TP (g kg^−1^)	TK (g kg^−1^)	AP (mg kg^−1^)	AK (mg kg^−1^)
2020–2021	8.24	25.67	1.18	1.07	18.21	11.20	197.22
2021–2022	8.50	26.33	1.02	1.03	16.34	13.38	196.65

## Data Availability

The data that support the findings of this study are available from the corresponding author upon reasonable request.

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
