# Peer review of "Mulched Drip Fertigation with Growth Inhibitors Reduces Bundle-Sheath Cell Leakage and Improves Photosynthesis Capacity and Barley Production in Semi-Arid Regions"

_plants, 2024, doi:10.3390/plants13020239_

Round 1
Reviewer 1 Report
Comments and Suggestions for Authors
the authors worked about Effect of mulch drip fertigation with growth inhibitors applications to reduce bundle sheath cell leakage, improves photosynthesis capacity and barley productivity in semi-arid regions.
The barley plants were exposed to novel nitrification inhibitor (NI) (3,4-dimethyl-1H-pyrazol-1-yl succinic acid) (DMPSA) and urease inhibitor (UI) N-butyl thiophosphorictriamide (NBPT) with mulch drip fertigation treatments which includes HF (high drip fertigation (370 mm) under ridge furrow system), MF (75 % of HF, moderate drip fertigation under ridge furrow system), LF (50 % of HF, low drip fertigation under ridge furrow system), and TP: (traditional planting with no inhibitors and drip fertigation strategies)
Under both inhibitors, there was a significant difference between HF and LF drip fertigation. Under the NI-MF treatment significantly increased grain yield, total Chl. ab content, WUE and 28 NUE by 52%, 47%, 57%, and 45%, respectively. Collectively, the results suggest that the new nitrification inhibitor (DMPSA) with HF or MF mulch drip fertigation could be promoted in semi-arid regions in order to mitigate bundle-sheath cell leakage to CO2 (ɸ) without penalizing barley production and leading to nutrient and water use efficiency of barley. the idea is good specially with new methods
but i still have some comments
the title is too long plz change it
the introduction
i suggest you read these ref
1- Use of nanoparticles in improving photosynthesis in crop plants under stress.SM Elhefnawy, NI Elsheery.Photosynthesis, 105-135
2-5-Aminolevulinic acid and 24-epibrassinolide improve the drought stress resilience and productivity of banana plants.MN Helaly, HM El-Hoseiny, NI Elsheery, HM Kalaji, ...Plants 11 (6), 743
3-Over expression of Jatropha’s Dehydrin Jcdhn-2 enhances tolerance to water stress in Rice plants.SA Omar, NI Elsheery, AA Elzaawely, W Strobel, H MKalaji.International Journal of Biosciences 13 (2), 53-60
4-Regulation and physiological role of silicon in alleviating drought stress of mango.MN Helaly, H El-Hoseiny, NI El-Sheery, A Rastogi, HM Kalaji.Plant physiology and biochemistry 118, 31-44
5-Gas exchange, chlorophyll fluorescence, and osmotic adjustment in two mango cultivars under drought stress.NI Elsheery, KF Cao.Acta Physiologiae Plantarum 30, 769-777
the aim of your study is not clear
M&M
make your method for light curve clear
Results
why you didnt measure Fv/Fm paramters
Fig 2 check your statistic i doubt about it
Your discussion part need to rewrite again
make connections between your parts
Comments on the Quality of English Language
its ok
Author Response
Thank you to review our manuscript! You are kind and responsible reviewer, and the suggestions you have given are all valuable and very helpful for revising and improving our paper. We are very grateful to that. We have studied your comments carefully and have made correction; Many thanks to the Editor and the Reviewers for your time and thoughtful comments, many of which have been incorporated into the revised manuscript.
Below are our detailed responses (in BOLD type) to Editor and Reviewers comments, (the page and line numbers refer to our revised manuscript):
The authors worked about Effect of mulch drip fertigation with growth inhibitors applications to reduce bundle sheath cell leakage, improves photosynthesis capacity and barley productivity in semi-arid regions.
The barley plants were exposed to novel nitrification inhibitor (NI) (3,4-dimethyl-1H-pyrazol-1-yl succinic acid) (DMPSA) and urease inhibitor (UI) N-butyl thiophosphorictriamide (NBPT) with mulch drip fertigation treatments which includes HF (high drip fertigation (370 mm) under ridge furrow system), MF (75 % of HF, moderate drip fertigation under ridge furrow system), LF (50 % of HF, low drip fertigation under ridge furrow system), and TP: (traditional planting with no inhibitors and drip fertigation strategies) under both inhibitors, there was a significant difference between HF and LF drip fertigation. Under the NI-MF treatment significantly increased grain yield, total Chl. ab content, WUE and 28 NUE by 52%, 47%, 57%, and 45%, respectively. Collectively, the results suggest that the new nitrification inhibitor (DMPSA) with HF or MF mulch drip fertigation could be promoted in semi-arid regions in order to mitigate bundle-sheath cell leakage to CO2 (ɸ) without penalizing barley production and leading to nutrient and water use efficiency of barley. The idea is good especially with new methods but i still have some comments
Response: respected reviewer, thank you very much for your encouragement and excellent suggestions. Sir I revise my article according to your below suggestions and comments; I hope the current version of my article will be acceptable for publication in PLANT MDPI Journal.
The title is too long plz change it
Response: Thank you for your advice, sir according to your suggestion I rewrite and modify the title of my article according to your above suggestion, please see in new revise paper.
The introduction
i suggest you read these ref
1- Use of nanoparticles in improving photosynthesis in crop plants under stress.SM Elhefnawy, NI Elsheery.Photosynthesis, 105-135
2-5-Aminolevulinic acid and 24-epibrassinolide improve the drought stress resilience and productivity of banana plants.MN Helaly, HM El-Hoseiny, NI Elsheery, HM Kalaji, ...Plants 11 (6), 743
3-Over expression of Jatropha’s Dehydrin Jcdhn-2 enhances tolerance to water stress in Rice plants.SA Omar, NI Elsheery, AA Elzaawely, W Strobel, H MKalaji.International Journal of Biosciences 13 (2), 53-60
4-Regulation and physiological role of silicon in alleviating drought stress of mango.MN Helaly, H El-Hoseiny, NI El-Sheery, A Rastogi, HM Kalaji.Plant physiology and biochemistry 118, 31-44
5-Gas exchange, chlorophyll fluorescence, and osmotic adjustment in two mango cultivars under drought stress.NI Elsheery, KF Cao.Acta Physiologiae Plantarum 30, 769-777
Response: Thank you for your advice, sir according to your above suggest articles I read and take a great help to improve the introduction section of my article, also sir I cited the above articles, sir please see in a new revise article.
The aim of your study is not clear
Response: We are sorry about that, sir according to your above suggestion I rewrite, and improve the last part (objectives / aims) of my introduction section with your above suggestion, sir please see in a new revise article.
M&M
Make your method for light curve clear
Response: We are sorry about that, sir I mention the detail method for light response curve in material and method section, please see in a new revise article.
Results
Why you didn’t measure Fv/Fm parameters
Response: Thank you for your suggestion, Sir I already measure 18 parameters in my article, if I add Fv/Fm parameters them the article will be too much lengthy that is why I did not add Fv/Fm parameters, also our article mainly focus to reduce bundle sheath cell leakage, improves photosynthesis capacity and barley productivity in semi-arid regions.
Fig 2 check your statistic I doubt about it
Response: Thank you, following your suggestion, Sir I recheck and re-statistical analysis and correct it, sir please see a revise Fig. 2 in a new revise article.
Your discussion part needs to rewrite again
Response: Thank you for your advice, sir according to your suggestion I rewrite and modify the discussion part with latest literature and cited the latest literature according to your above suggestion, please see in new revise paper.
Make connections between your parts
Response: We are sorry about that, sir according to your above suggestion I make a strong connections between the result and discussion part of my article, sir please sees in a new revise article.

Reviewer 2 Report
Comments and Suggestions for Authors
A brief summary
The manuscript “Effect of mulch drip fertigation with growth inhibitors applications to reduce bundle sheath cell leakage, improves photosynthesis capacity and barley productivity in semi-arid regions” is a study case about reducing bundle-sheath cell leakage and improving photosynthesis in barley.
General concept comments
Barley plants treated with nitrification and urease inhibitors are important for farmers with use mulch drip fertigation system. The paper shows that inhibitors-based strategy reduces bundle-sheath cell leakage by 35-39%. Also, it highlights that productivity and photosynthesis capacity is improved under those inhibitors which are recommended in semi-arid regions.
In this form the manuscript could not be suitable for publication in Plants journal, because there are major shortcomings of describe clarity and analyses method. I recommend a major revision of manuscript.
Specific comments
My comments regarding the aspect which needs to be major revision:
1) In manuscript, it is presented ”new types of nitrification and urease inhibitors” (page 2 line 67) and this idee is repeated more times. I identified only nitrification inhibitor (3,4-dimethyl-1H-pyrazol-1-yl succinic 13 acid) and urease inhibitor (N-butyl thiophosphorictriamide), which are already used in agriculture. Which are the new types of inhibitors?
2) The title isn't cover complete in sections of manuscript. The barley productivity is lack in article. An economic section should be included in order to show how productivity is increasing by using each type of inhibitor on the hectare cultivated.
3) There are lot of recent studies which describe the role of nitrification and urease inhibitors used in this study. The discussion section should be extended by consider the newest research conducted by other groups.
4) In tables 2-5, the a superscript is used to describe the treatment efficient and b indicate significant differences. This coincidence of letters is difficult to understand. In general, the a-e letters are used for describing the significant differences between the values according to statistical test. I recommend change the notations and, more importantly, manuscript must be improved by describe the significance of statistical analysis.
5) The results are described in detail with tables, figures and values but without indicating the importance of each method regarding the purpose of the article.
6) In text, a lot of abbreviations are used before to be explained. For example, ET rate (page 2 line 67), PEP (page 2 line 81), NUE (page 2 line 92, the complete name is at page 5 line 188), WUE (page 2 line 92 and complete name at page 10 line 318), Chl contents and others. To be easy to read, in manuscript must be used complete name of terms where aper first time.
Author Response
Thank you to review our manuscript! You are kind and responsible reviewer, and the suggestions you have given are all valuable and very helpful for revising and improving our paper. We are very grateful to that. We have studied your comments carefully and have made correction; Many thanks to the Editor and the Reviewers for your time and thoughtful comments, many of which have been incorporated into the revised manuscript.
Below are our detailed responses (in BOLD type) to Editor and Reviewers comments, (the page and line numbers refer to our revised manuscript):
A brief summary
The manuscript “Effect of mulch drip fertigation with growth inhibitors applications to reduce bundle sheath cell leakage, improves photosynthesis capacity and barley productivity in semi-arid regions” is a study case about reducing bundle-sheath cell leakage and improving photosynthesis in barley.
Response: respectable reviewer, thank you very much for your encouragement and excellent suggestions. Sir I revise my article according to your below suggestions and comments; I hope the current version of my article will be acceptable for publication in PLANT MDPI Journal.
General concept comments
Barley plants treated with nitrification and urease inhibitors are important for farmers with use mulch drip fertigation system. The paper shows that inhibitors-based strategy reduces bundle-sheath cell leakage by 35-39%. Also, it highlights that productivity and photosynthesis capacity is improved under those inhibitors which are recommended in semi-arid regions.
Response: respectable reviewer, thank you very much for your encouragement and excellent suggestions.
In this form the manuscript could be suitable for publication in Plants journal after a major revision, because there are major shortcomings of describe clarity and analyses method. I recommend a major revision of manuscript.
Response: respectable reviewer, thank you very much for your excellent suggestions. Sir I revise my article according to your below suggestions and comments; I hope the new version of my article will be acceptable for publication.
Specific comments
My comments regarding the aspect which needs to be major revision:
- In manuscript, it is presented ”new types of nitrification and urease inhibitors” (page 2 line 69) and this idea is repeated more times. I identified only nitrification inhibitor (3,4-dimethyl-1H-pyrazol-1-yl succinic 13 acid) and urease inhibitor (N-butyl thiophosphorictriamide), which are already used in agriculture. Which are the new types of inhibitors?
Response: Thank you for your comment, sir I mean that nitrification and urease inhibitors are use first time with combination of mulch drip fertigation that is why I say new types, but sir according to your suggestion I delete the new types in the above sentence, sir please see in an new revise article.
- The title isn't cover complete in sections of manuscript. An economic section should be included in order to show how productivity is increasing by using each type of inhibitor on the hectare cultivated.
Response: We are sorry about that, sir according to your above suggestion I included the economic analysis section according to your above suggestion, sir please see in a new revise article (Table 6).
- There are lot of recent studies which describe the role of nitrification and urease inhibitors used in this study. The discussion section should be extended by consider the newest research conducted by other groups.
Response: Thank you for your advice, sir according to your suggestion I read a recent articles which describe the role of nitrification and urease inhibitors used in this study, according to the latest articles I rewrite and modify the discussion part with latest literature and cited the latest literature according to your above suggestion, please see in new revise paper.
- In tables 2-5, the a superscript is used to describe the treatment efficient and b indicate significant differences. This coincidence of letters is difficult to understand. In general, the a-e letters are used for describing the significant differences between the values according to statistical test. I recommend change the notations and, more importantly, manuscript must be improved by describe the significance of statistical analysis.
Response: We are sorry about that, sir I change the notations according to your suggestion and describe the significance of statistical analysis, please see in a new revise article.
- The results are described in detail with tables, figures and values but with indicating the importance of each method regarding the purpose of the article.
- Response: respectable reviewer, thank you very much for your encouragement and excellent suggestions.
- In text, a lot of abbreviations are used before to be explained. For example, ET rate (page 2 line 67), PEP (page 2 line 81), NUE (page 2 line 92, the complete name is at page 5 line 188), WUE (page 2 line 92 and complete name at page 10 line 318), Chl contents and others. To be easy to read, in manuscript must be used complete name of terms where a per first time.
Response: We are sorry about that, sir according to your above suggestion I explained the first appearance of abbreviations in my whole article, sir please sees in a new revise article.

Reviewer 3 Report
Comments and Suggestions for Authors
In this study, the authors examined the effects of nitrification and urease inhibitors combined with mulch irrigation system on physiological parameters and yield of barely crop. The topic of this article is very interesting; however, the practices of mulching and drip fertigation, usually are not applied in cereals low to high cost of installation. Also, in this study the row spacing is too wide (60 cm) for this crop. Usually, the row spacing in winter cereals cultivation is about 20 cm. Additionally, mulching is usually applied in summer crops (e.g., vegetable crops) and not in winter crops. In the introduction section the authors they didn’t show the significance of two methods (mulching and drip fertigation) for barley or other winter cereals crops, while in the discussion section the authors should compare their results with the results of other studies. Thus, in my opinion this article should be rejected, although the authors measured several parameters of barley crop.
Author Response
Thank you to review our manuscript! You are kind and responsible reviewer, and the suggestions you have given are all valuable and very helpful for revising and improving our paper. We are very grateful to that. We have studied your comments carefully and have made correction; Many thanks to the Editor and the Reviewers for your time and thoughtful comments, many of which have been incorporated into the revised manuscript.
Below are our detailed responses (in BOLD type) to Editor and Reviewers comments, (the page and line numbers refer to our revised manuscript):
In this study, the authors examined the effects of nitrification and urease inhibitors combined with mulch irrigation system on physiological parameters and yield of barely crop. The topic of this article is very interesting; however, the practices of mulching and drip fertigation usually are applied in cereals low to high cost of installation.
Response: respected reviewer, thank you very much for your encouragement and excellent suggestions. Sir I revise my article according to your below suggestions and comments; I hope the current version of my article will be acceptable for publication in PLANT MDPI Journal.
Also, in this study the row spacing is too wide (60 cm) for this crop. Usually, the row spacing in winter cereals cultivation is about 20 cm.
Response: We are sorry about that, Sir really it’s a typing mistake that we write row space 60 cm sir in our experiment have also 20 cm row space, sir you are right that row space is 20 cm, sir I check and correct the row space 20 cm, please see in a new revise article.
Additionally; mulching is usually applied in summer crops (e.g., vegetable crops) and not in winter crops.
Response: respected reviewer, we conduct our experiment under the dry-land farming system with no concept of irrigation with rainfall in 2020-21 was (230 mm), and in 2021-22 it was (200 mm) and completely depend on rainfall water which is not enough for barley growth that is why we use mulching to efficiently use rainfall water with drip fertigation.
In the introduction section the authors they didn’t show the significance of two methods (mulching and drip fertigation) for barley or other winter cereals crops, while in the discussion section the authors should compare their results with the results of other studies. Thus, in my opinion this article should be rejected and resubmit, although the authors measured several parameters of barley crop.
Response: Thank you for your advice, sir according to your suggestion I rewrite and modify introduction show the significance of two methods (mulching and drip fertigation) for barley, also rewrite and modify discussion part of my article with latest literature and cited the latest literature according to your above suggestion, please see in new revise paper. Sir I revise my article according to your below suggestions and comments; I hope the current version of my article will be acceptable for publication in PLANT MDPI Journal.

Round 2
Reviewer 1 Report
Comments and Suggestions for Authors
the authors did all suggestion which i asked
Comments on the Quality of English Languageits ok
Author Response
The authors did all suggestion which i asked
Response: respected reviewer, thank you very much for your encouragement and excellent suggestions. I hope the current version of my article will be acceptable for publication in PLANT MDPI Journal.
Reviewer 2 Report
Comments and Suggestions for Authors
Dear authors,
I appreciate your response. You sent detailed explanations at my comments. After a complete revision, the manuscript has been improved by including more explanations, so in this form is clear and easy to read.
Finally, I have a last comment:
- The figures have a low resolution and some of them are distorted because the length-to-width ratio was not respected (the most distorted is figure 5)
Author Response
Dear authors,
I appreciate your response. You sent detailed explanations at my comments. After a complete revision, the manuscript has been improved by including more explanations, so in this form is clear and easy to read.
Response: respectable reviewer, thank you very much for your encouragement and excellent suggestions. I hope the current version of my article will be acceptable for publication in PLANT MDPI Journal.
Finally, I have a last comment:
- The figures have a low resolution and some of them are distorted because the length-to-width ratio was not respected (the most distorted is figure 5)
Response: Thank you for your advice, sir according to your suggestion I improve the figures resolution especially Figure 5, sir please see in a new revise article.
Reviewer 3 Report
Comments and Suggestions for Authors
Dear Editor,
The authors during the review process improved the article. Moreover, the authors have adequately addressedof my comments and thus, the article can be accepted for puplication in Plants after minor revision.
Comments
* Matrerial and methods section should be moved after the discussion section.
**References should be corrected according to instruction for authors.
Best Regards,
Anestis Karkanis
Author Response
The authors during the review process improved the article. Moreover, the authors have adequately addressed of my comments and thus, the article can be accepted for publication in Plants after minor revision.
Response: respected reviewer, thank you very much for your encouragement and excellent suggestions. Sir I revise my article according to your below suggestions and comments; I hope the current version of my article will be acceptable for publication in PLANT MDPI Journal.
Comments
* Material and methods section should be moved after the discussion section.
Response: Thank you for your advice, sir according to your suggestion I move the material and methods section after discussion, please see in new revise paper.
**References should be corrected according to instruction for authors.
Response: Thank you for your advice, sir according to your suggestion I correct the references according to journal format and instruction, please see in new revise paper.